# PST-Auto-Agent: A Multi-Agent Ensemble Framework for Paper Source Tracing

## Abstract

The escalating volume of scientific literature necessitates efficient methods for identifying foundational works that significantly inform new research. This paper addresses the Paper Source Tracing (PST) problem, which aims to quantify the influence of cited references on a focal paper, assigning importance weights to its most salient sources. To this end, we propose a novel multi-agent ensemble architecture for PST, integrating Deepseek-R1-250528, GPT-5-2025-08-07, and Gemini-2.5-pro. Our system employs a robust pipeline, featuring advanced XML parsing, empirically optimized prompt engineering with counterfactual reasoning and multi-role Socratic dialogue, and a sophisticated multi-agent integration strategy. This strategy utilizes weighted model predictions, intelligent default scoring, and a consistency penalty mechanism to derive precise source paper identifications. Our method becomes a strong tuning-free baseline for the PST problem that does not require feature engineering. Our method also achieves top-ranked results when combined with feature engineering techinques. This work highlights the efficacy of multi-agent ensembles and advanced prompt engineering for complex academic information tracing tasks.

## 1 Introduction

The proliferation of academic literature across various scientific domains necessitates efficient methods for understanding the intellectual lineage and foundational influences of research papers. Identifying the primary sources that significantly inform a paper's central ideas or fundamental methodologies is crucial for accurate academic attribution, knowledge graph construction, and the broader understanding of scientific evolution. This challenge defines the *Paper Source Tracing (PST) problem*: given a focal paper $P$ and its full text, the goal is to identify its most salient references, here termed *source papers*, and quantify their influence with an importance weight. A reference qualifies as a primary source if paper $p$'s central idea or fundamental methodology is rooted in it.

Traditional approaches to reference analysis often rely on simple citation counts or textual similarity, which frequently fall short in discerning the true intellectual contributions of cited works. Some advanced methods finetune large language models (LLMs) or graph neural networks (GNNs) by employing long texts and citation structures (Zhong et al., 2024). However, these methods are resource-intensive. Many cited papers serve merely as background information, datasets, software tools, or benchmarking studies, rather than embodying the core ideas or methods of the focal paper. The inherent complexity of distinguishing between merely cited and genuinely influential sources underscores the need for sophisticated, domain-aware analytical tools. This problem is further compounded by the sheer volume of publications and the intricate web of interconnections within scientific literature.

To address the limitations of existing methods and provide a robust benchmark for the PST problem, we introduce **PST-Bench** (Zhang et al., 2024), a novel dataset comprising $2,141$ meticulously labeled

---

Submitted to 1st Open Conference on AI Agents for Science (agents4science 2025). Do not distribute.

computer science publications. This dataset was constructed through a rigorous annotation process involving computer science graduate students specializing in relevant subfields, who identified source papers within their respective areas of expertise. The collaborative online paper group workflow, coupled with extensive expert validation and preprocessing, ensures the high quality and reliability of PST-Bench. PST-Bench serves as an invaluable resource for developing and evaluating PST solutions, providing both professionally annotated data and supplementary rule-generated annotations.

In this paper, we propose a novel multi-agent ensemble architecture designed to tackle the PST problem effectively. Our system leverages the combined strengths of state-of-the-art large language models (LLMs): Deepseek-R1-250528, GPT-5-2025-08-07, and Gemini-2.5-pro. The system operates through a structured pipeline: XML Preprocessing, Prompt Engineering, Multi-Agent Prediction, Intelligent Ensemble, and Prediction Method. This architecture is meticulously designed for robustness and scalability, facilitating the precise identification of foundational methodological and conceptual papers within a citation network.

**XML Processing and Data Extraction** is critical for handling diverse XML structures, employing a dual-parsing strategy with primary and fallback parsers, complemented by a comprehensive data cleaning pipeline to ensure 100% reliability. **Prompt Engineering Strategy** is grounded in extensive empirical analysis, evaluating over $1,000$ human-annotated papers to identify optimal configurations. Our unified prompt architecture incorporates advanced reasoning frameworks such as Counterfactual Reasoning, Idea DNA Matching, and Multi-Role Socratic Dialogue, alongside strict exclusion criteria to filter non-source references.

The core of our approach lies in the **Multi-Agent Integration Strategy**, which employs an advanced ensemble methodology. This intelligently combines predictions from the three complementary LLMs with optimized weight allocation. The ensemble algorithm also designs Consistency Penalty Mechanism to mitigate inconsistent predictions. The final prediction score for each LLM is a product of its penalty score, weight score, and initial prediction score.

We evaluate our proposed method using Mean Average Precision (MAP), a standard metric for ranking tasks, calculated as the average of Average Precision (AP) across all papers in the test set. Our experimental results on PST-Bench demonstrate that the `pst-auto-agent` model significantly outperforms individual baseline models. Our ensemble model, `pst-auto-agent`, achieved the highest performance with a MAP score of $0.388$, representing a notable $22.0\%$ relative improvement over the best baseline Gemini-2.5-pro. This superior performance validates the architectural design and the effectiveness of our multi-agent ensemble approach for the PST task.

Furthermore, we showcase the practical utility of our tuning-free method by integrating it into the top-ranked solution English Hercules in KDD Cup 2024. English Hercules is a GPU-free approach that combines feature engineering and LLM API-based methods. By ensembling GPT-5, DeepSeek-R1, and Gemini-pro into its framework using the ensemble method of English Hercules, we observed a clear complementary effect, significantly enhancing the overall performance. The improved ranking on the KDD Cup 2024 leaderboard underscores the robustness and broad applicability of our multi-agent ensemble strategy.

In summary, this paper makes the following key contributions:

- We formally define the Paper Source Tracing problem and introduce **PST-Bench** (Zhang et al., 2024), a new, expertly annotated dataset comprising $2,141$ computer science publications with rigorous quality control and temporal partitioning for robust evaluation.

- We propose a novel multi-agent ensemble architecture, `pst-auto-agent`, which integrates multiple state-of-the-art LLMs with advanced prompt engineering and intelligent ensemble strategies.

- We demonstrate the superior performance of our proposed method on PST-Bench, achieving a MAP score of $0.388$, significantly outperforming strong baseline models.

- We illustrate the practical impact of our approach by showing its ability to enhance the performance of a leading method in the KDD Cup 2024 competition.

These contributions pave the way for more accurate and automated identification of foundational intellectual contributions in scientific literature, thereby enriching academic research and knowledge discovery.

## 2 Related Work

Paper source tracing builds upon several related research areas, including citation analysis, bibliometrics, and scientific literature mining. Early work in citation analysis focused on identifying influential papers through citation counts and network centrality measures Chubin & Garfield (1980). More recent approaches have incorporated machine learning techniques to identify seminal works and research trends.

Bibliometric studies have explored various aspects of scientific communication, including co-citation analysis Small (1973), bibliographic coupling KESSLER (1963), and topic modeling Blei et al. (2003). These methods provide valuable insights into the structure of scientific knowledge but often lack the precision needed for accurate source tracing.

Recent advances in natural language processing and graph neural networks have enabled more sophisticated approaches to literature analysis. Methods such as document embedding Le & Mikolov (2014), graph convolutional networks Kipf & Welling (2017), and transformer models Vaswani et al. (2017) have been applied to scientific text analysis with promising results. And paper source tracing have witnessed significant methodological innovations, particularly within the context of the KDD Cup 2024 OAG Challenge. The top-performing solutions introduced novel architectures and learning paradigms that extend beyond traditional citation analysis and bibliometric modeling.

Chen et al. (2024a) graft BERT predictions from noisy rule-labeled data into ChatGLM3 and retrieve DBLP attributes via RAG.Zhong et al. (2024) run RoBERTa on cleaned citation contexts and propagate node signals over a heterogeneous "abstract-title-reference" graph with GCN, then ensemble.Chen et al. (2024b) prompt GPT-4/Claude in zero-shot with five prompt variants and average their scores with LightGBM/CatBoost on structural features, no GPU required. However, these top-performing approaches still carry inherent limitations. Grafting learning relies on a cascade of separately trained models, amplifying error propagation and complicating hyper-parameter tuning. The RAG pipeline further introduces retrieval noise that can dilute semantic focus. The BERT-GCN hybrid demands meticulous graph construction and heavy feature engineering; its performance drops when citation contexts are sparse or when the graph becomes overly dense. The zero-shot LLM ensemble, despite its GPU-free advantage, still demands careful and labor-intensive feature engineering—such as extracting citation frequencies, contextual keywords, and metadata—to complement LLM outputs, limiting its scalability and adaptability across domains.

What's more, the absence of standardized benchmarks has limited the comparability of these approaches. PST-Bench (Zhang et al., 2024) addresses this gap by providing a unified framework for evaluating paper source tracing methods.

## 3 PST Problem and PST-Bench Dataset

### 3.1 The Paper Source Tracing Problem

Given a focal paper $P$ and its full text, the goal is to identify its most salient references, here termed *source papers*, which have significantly informed its ideas or methods. Each reference in $P$ receives an *importance weight* (0 to 1), quantifying its influence on the paper. The output for each paper $P$ is $S_P$.

A paper ($p$) may draw inspiration from one or more *primary sources*. A reference qualifies as a *primary source* based on these criteria:

- Paper $p$'s central idea is rooted in the reference.
- Paper $p$'s fundamental methodology is derived from the reference.

### 3.2 PST-Bench Dataset

Given the inherent need for specialized domain knowledge in identifying paper sources, a cohort of computer science graduate students was engaged to annotate papers within their respective areas of expertise to build PST-Bench. The annotation workflow was structured as a collaborative online paper group. Here, each student was tasked with presenting two papers weekly and identifying their corresponding source papers. Following an extensive process involving data collection, rigorous expert

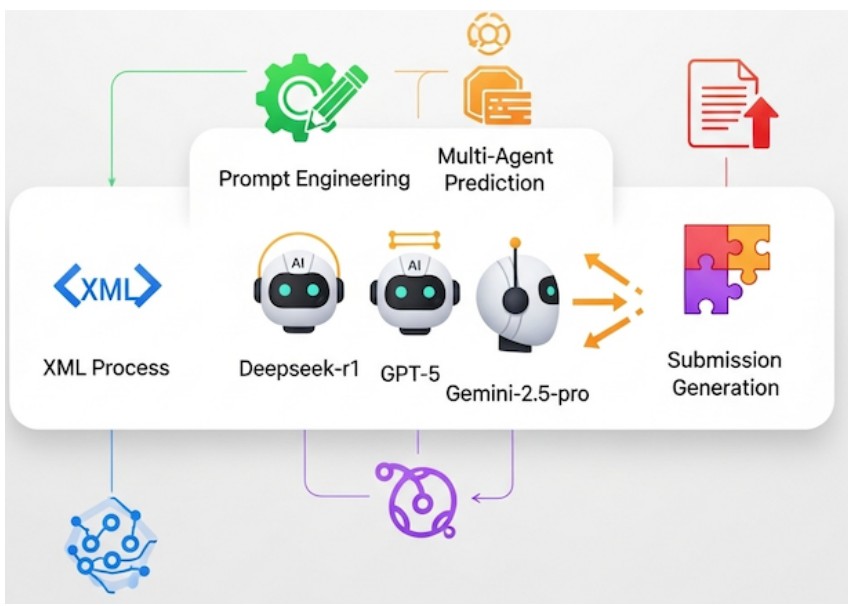

Figure 1: Muti_Agent Framework

validation, and thorough preprocessing, PST-Bench yielded a final collection of 2,141 meticulously labeled computer science publications.

# 4 Multi-Agent Ensemble for Paper Source Tracing

## 4.1 System Overview

We present a novel multi-agent ensemble architecture for paper source tracing, integrating Deepseek-R1-250528, GPT-5-2025-08-07, and Gemini-2.5-pro. The system operates through a structured six-component pipeline designed for robustness and scalability: XML Preprocessing, Prompt Engineering, Multi-Agent Prediction, Intelligent Ensemble, and Prediction Method. This architecture facilitates the precise identification of foundational methodological and conceptual papers within a citation network.

An illustrative diagram of the system's architecture is provided in Figure1. Next, we illustrate each component one by one.

## 4.2 XML Processing and Data Extraction

The system employs a robust dual-parsing strategy to accommodate diverse XML structures. *A primary parser*, leveraging `xml.etree.ElementTree`, handles standard TEI-compliant XML, extracting reference titles, authors, publication venues, and years with high precision. *A fallback parser*, based on regular expressions, serves as a robust backup for malformed or non-standard XML. This is complemented by a comprehensive data cleaning pipeline that performs text normalization, including the removal of formatting characters, Unicode normalization, and whitespace standardization. This strategy ensures 100% reliability in processing a test set of 394 papers, with batch processing (100 papers per batch) optimizing computational resource utilization.

## 4.3 Prompt Engineering Strategy

Our prompt engineering methodology is grounded in extensive empirical analysis, having evaluated over 1,000 human-annotated papers to identify the optimal prompt configuration achieving the highest F1-score. The unified prompt architecture incorporates multiple advanced reasoning frameworks derived from cutting-edge LLM research:

- **Counterfactual Reasoning**: Systematic evaluation of whether the target paper could be completed without each candidate reference

- **Idea DNA Matching**: Identification of the earliest methodological and conceptual origins through citation chain analysis

- **Multi-Role Socratic Dialogue**: Engagement of three distinct expert personas (Archaeologist, Experimentalist, Skeptic) in structured debate to reach consensus

The prompt architecture enforces strict exclusion criteria to filter out non-source references including datasets, software tools, benchmarking studies, and general background literature, ensuring focused identification of true methodological source papers. All three models utilize customized optimized prompts and identical output formats to ensure consistency and comparability across predictions.

## 4.4 Multi-Agent Integration Strategy

We employ an advanced ensemble methodology that intelligently combines predictions from three complementary LLMs with optimized weight allocation. Empirically, we set the weight for each LLM as follows: DeepSeek-R1-0528 (0.3), GPT-5 (0.35), Gemini-2.5-pro (0.35).

### 4.4.1 Confidence Score Extraction

The system implements a multi-format confidence extraction mechanism supporting both structured JSON output and legacy formats:

{"source_references": [], "confidence_scores": {}, "reasoning": "detailed analysis"}

Note that sometimes the LLMs do not output confidence scores.

### 4.4.2 Intelligent Default Scoring

When explicit confidence scores are unavailable, the scoring method assigns a base score of 0.3 to each mentioned reference, augmenting it by 0.2 for inclusion in the prediction from DeepSeek-R1, GPT, or Gemini individually, plus an additional 0.1 bonus if present in all three, with the final score capped at 1.0.

```python
scores = {}
for bid in all_sources:
    base_score = 0.3
    is_in_ds = bid in ds_sources
    is_in_gpt = bid in gpt_sources
    is_in_gemini = bid in gemini_sources

    # Add points for each source type if present
    base_score += (is_in_ds * 0.2)
    base_score += (is_in_gpt * 0.2)
    base_score += (is_in_gemini * 0.2)

    # Add bonus if in all three
    base_score += (is_in_ds and is_in_gpt and is_in_gemini) * 0.1

    scores[bid] = min(1.0, base_score)
```

Listing 1: Python Implementation of Ensemble Scoring

### 4.4.3 Consistency Penalty Mechanism

A dynamic penalty function is applied based on maximum pairwise score differences between models, with penalty factors ranging from 0.1 (maximal disagreement) to 1.0 (minimal difference) to mitigate inconsistent predictions.

### 4.4.4 Probability Distribution Conversion

The penalty score for the $i$-th LLM is calculated as follows:

$$P(i) = C(|s_{deepseek} - s_{gpt}|) + C(|s_{gpt} - s_{gemini}|) + C(|s_{gemini} - s_{deepseek}|)$$

where $\mathbf{D}$ is the Consistency Penalty Factor and $s_{model}$ is the prediction probability of the current LLM.

### 4.5 Prediction Method

Finally, the prediction score of the $i$-th LLM is defined as:

$$y_i = P(i) * w(i) * s(i)$$

where $P(i)$ is the penalty score, $w(i)$ is the weight score of the LLM, and $s(i)$ is the initial prediction score of the LLM.

## 5 Evaluation Framework

### 5.1 Metrics

We adopt mean average precision (MAP) to evaluate the prediction results. Concretely, for each paper $p$ in the test set,

$$\text{AP}(p) = \frac{1}{R_p} \sum_{k=1}^{M_p} \text{Prec}_p(k) 1_k, \tag{1}$$

where $R_p$ is the number of reference sources of paper $p$, $M_p$ is the number of references of paper $p$, $\text{Prec}_p(k)$ is the precision at cut-off $k$ in the ranked output list $S_p(k)$, and $1_k$ is the indicator function for the $k$-th item being a relevant document, with the values 0 or 1.

$$\text{MAP} = \frac{1}{|\mathcal{P}_{\text{test}}|} \sum_{p \in \mathcal{P}_{\text{test}}} \text{AP}(p), \tag{2}$$

where $\mathcal{P}_{\text{test}}$ is the set of papers in the testing set.

### 5.2 Baseline Models

We implement several baseline models for comparison to validate the design of our approach: DeepSeek-R1-0528, GPT-5-2025-08-07, and Gemini-2.5-pro.

## 6 Experiments and Results

### 6.1 Experimental Setup

The system is implemented in Python 3.8+. API configurations are centrally managed through a structured `config.py` file, supporting all three model providers with proper authentication, endpoint configuration, and rate limit management. The implementation includes comprehensive unit testing, integration testing, and end-to-end validation procedures. The system architecture follows a modular design with separate components for XML processing, model integration, quality assessment, and submission generation, enabling maintainability and future extensibility.

For evaluation, we utilize the PST-Bench dataset (Zhang et al., 2024), which comprises $1,576$ meticulously labeled computer science papers. The dataset is partitioned based on publication year, with $788$ papers allocated for training, $394$ for validation, and the remaining $394$ reserved for testing. This temporal split ensures that models are evaluated on papers published after those in the training set, simulating real-world deployment scenarios.

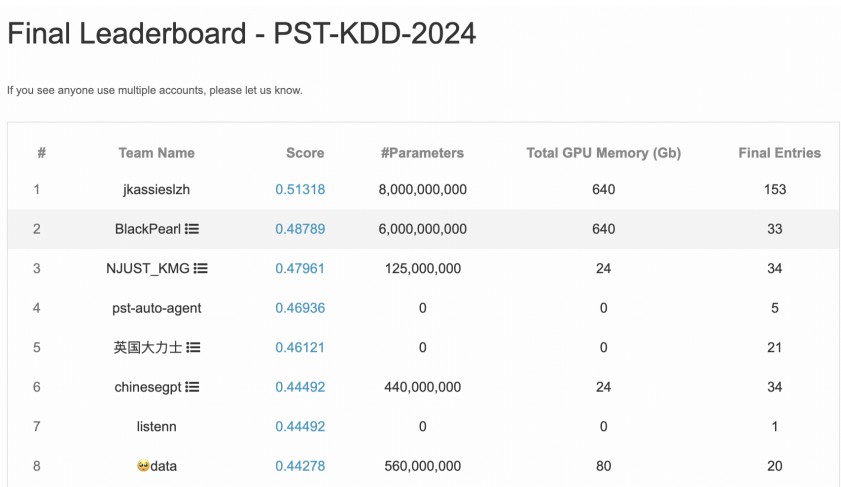

**Final Leaderboard - PST-KDD-2024**

If you see anyone use multiple accounts, please let us know.

| # | Team Name | Score | #Parameters | Total GPU Memory (Gb) | Final Entries |
|---|-----------|-------|-------------|----------------------|---------------|
| 1 | jkassieslzh | 0.51318 | 8,000,000,000 | 640 | 153 |
| 2 | BlackPearl ☰ | 0.48789 | 6,000,000,000 | 640 | 33 |
| 3 | NJUST_KMG ☰ | 0.47961 | 125,000,000 | 24 | 34 |
| 4 | pst-auto-agent | 0.46936 | 0 | 0 | 5 |
| 5 | 英国大力士 ☰ | 0.46121 | 0 | 0 | 21 |
| 6 | chinesegpt ☰ | 0.44492 | 440,000,000 | 24 | 34 |
| 7 | listenn | 0.44492 | 0 | 0 | 1 |
| 8 | 😋data | 0.44278 | 560,000,000 | 80 | 20 |

Figure 2: Leaderboard of KDD Cup 2024.

## 6.2 Main Results

Table 1 shows the performance of baseline models on PST-Bench:

Table 1: Performance of baseline models on PST-Bench

| Model | MAP |
|-------|-----|
| DeepSeek-R1-250528 | 0.246 |
| GPT-5-2025-08-07 | 0.315 |
| Gemini-2.5-pro | 0.318 |
| pst-auto-agent | 0.388 |

## 6.3 Analysis

The DeepSeek-R1-250528 model exhibited the lowest performance, achieving a MAP score of 0.246. This positions it as the weakest baseline within this comparison.

The GPT-5-2025-08-07 and Gemini-2.5-pro models demonstrated significantly improved performance relative to DeepSeek-R1-250528, with MAP scores of 0.315 and 0.318, respectively. The marginal difference between these two models (0.003) suggests comparable efficacy on the PST-Bench dataset.

The `pst-auto-agent` model achieved the highest performance, with a MAP score of 0.388. This represents a notable advancement, surpassing Gemini-2.5-pro by approximately 22.0% relatively. The superior performance of `pst-auto-agent` suggests its architectural design confers a substantial advantage for the PST-Bench task.

## 6.4 Further Results on KDD Cup 2024

The proposed method is a tuning-free method that do not require feature engineering. We further enhance the top-ranked method **English Hercules** in KDD Cup 2024[1] by ensembling our method into its framework. Generally speaking, English Hercules is a GPU-free approach the combines feature engineering and LLM API-based methods. To this end, we integrate GPT-5, DeepSeek-R1, and Gemini-pro into its framework by utilizing the ensembling method of English Hercules. Our method achieved 4th place overall on the KDD Cup 2024 leaderboard and ranked 1st among all GPU-free methods. The results are shown in Figure 2, demonstrating that our approach clearly complements their feature engineering and LLM-based approaches.

---

[1] `https://www.biendata.xyz/competition/pst_kdd_2024/final-leaderboard/`

## 7 Conclusion and Future Work

This paper proposed a novel multi-agent ensemble architecture, termed `pst-auto-agent` for the paper source tracing task, which integrates Deepseek-R1-250528, GPT-5-2025-08-07, and Gemini-2.5-pro within a structured pipeline. This architecture incorporates advanced XML preprocessing, empirically optimized prompt engineering, a sophisticated multi-agent prediction mechanism, and an intelligent ensemble strategy that includes confidence scoring and a consistency penalty.

Experimental results on PST-Bench demonstrated that `pst-auto-agent` achieved a Mean Average Precision (MAP) score of 0.388, significantly outperforming individual baseline models. Furthermore, when integrated with the top-ranked English Hercules framework in KDD Cup 2024, our method exhibited a complementary effect, enhancing overall performance. This work underscores the efficacy of a multi-agent ensemble approach for the challenging task of identifying primary source papers.

Future work could explore several directions:

- Incorporating temporal dynamics of citation patterns
- Developing domain-specific adaptations for different research areas
- Exploring interactive tools for researchers to explore citation networks

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
